# Robust Generation of Ready-to-Use Cryopreserved Motor Neurons from Human Pluripotent Stem Cells for Disease Modeling

**DOI:** 10.3390/ijms232113462

**Published:** 2022-11-03

**Authors:** Hsiao-Chien Ting, Hong-Lin Su, Mei-Fang Chen, Horng-Jyh Harn, Shinn-Zong Lin, Tzyy-Wen Chiou, Chia-Yu Chang

**Affiliations:** 1Bioinnovation Center, Buddhist Tzu Chi Medical Foundation, Hualien 970, Taiwan; 2Department of Life Sciences, National Chung Hsing University, Taichung 402, Taiwan; 3Department of Medical Research, Hualien Tzu Chi Hospital, Hualien 970, Taiwan; 4Department of Pathology, Hualien Tzu Chi Hospital and Tzu Chi University, Hualien 970, Taiwan; 5Department of Neurosurgery, Hualien Tzu Chi Hospital, Hualien 970, Taiwan; 6Department of Life Science and Graduate Institute of Biotechnology, National Dong Hwa University, Hualien 974, Taiwan; 7Neuroscience Center, Hualien Tzu Chi Hospital, Hualien 970, Taiwan; 8Department of Nursing, Tzu Chi University of Science and Technology, Hualien 970, Taiwan

**Keywords:** pluripotent stem cell, motor neuron, cryopreservation, electrophysiology, neuromuscular junction, disease modeling, amyotrophic lateral sclerosis, drug screening

## Abstract

Human pluripotent stem cell (hPSC)-derived motor neurons (MNs) act as models for motor neuron diseases (MNDs), such as amyotrophic lateral sclerosis (ALS) or spinal muscular atrophy. However, the MN differentiation efficiency and viability following cryopreservation require further development for application in large-scale studies and drug screening. Here, we developed a robust protocol to convert hPSCs into MN cryopreservation stocks (hPSCs were converted into >92% motor neural progenitors and >91% MNs). Near-mature MNs were cryopreserved at a high thawing survival rate and 89% MN marker expression on day 32. Moreover, these MNs exhibited classical electrophysiological properties and neuromuscular junction (NMJ) formation ability within only 4–6 days after thawing. To apply this platform as an MND model, MN stocks were generated from SOD1^G85R^, SOD1^G85G^ isogenic control, and sporadic ALS hPSC lines. The thawed ALS MNs expressed ALS-specific cytopathies, including SOD1 protein aggregation and TDP-43 redistribution. Thus, a stable and robust protocol was developed to generate ready-to-use cryopreserved MNs without further neuronal maturation processes for application in MND mechanistic studies, NMJ model establishment, and large-scale drug screening.

## 1. Introduction

Motor neurons (MNs) in the central nervous system (CNS), including the cerebral cortex (upper MNs), brain stem, and spinal cord (lower MNs), responsively modulate skeletal muscle contractions through MN–skeletal muscle junction structures known as neuromuscular junctions (NMJ), which produce body movements. MN loss or degeneration causes the detachment of NMJs, generating serious motor neuron diseases (MND), such as amyotrophic lateral sclerosis (ALS) and spinal muscular atrophy (SMA), engendering rapid motor function loss and patient death within a few years. However, effective treatment options for NMDs are yet to be established [1,2,3,4]. Pluripotent stem cells (PSCs), including human embryonic stem cells (hESCs) and induced pluripotent stem cells (iPSCs), are promising sources to obtain differentiated cell types suitable for human developmental and in vitro disease studies, especially as patient-derived cell models [5].

For MND models, patient-derived iPSCs need to be converted into MN progenitors (MNPs) and MNs to investigate MND-specific disease markers for detailed mechanistic studies or drug screening [6,7,8,9]. Thus, developing a rapid, stable protocol to convert PSCs into MNPs and MNs is the key to large-scale application of these models. Three steps are involved in differentiating PSCs into functional MNs: neural stem cell (NSC) induction, MNP patterning, and MN maturation.

Two mechanical protocols are widely used for NSC induction, monolayer-adherent and floating serum-free embryoid bodies (SFEB). The most widely used monolayer protocol was first developed in 2009 and it induced NSCs via dual SMAD inhibition [10]. Monolayer differentiation provides several advantages, such as rapid NSC conversion and easy cell morphology observation; nevertheless, it has technical challenges involving adherent conditions, cell confluence, and passage timing control that may affect efficiency during induction. The SFEB protocol was developed to convert PSCs into NSCs by activating FGF basic (FGFb) and activin inhibition [11,12]; however, previous reports suggested that this protocol requires longer time than the monolayer differentiation protocol to obtain NSCs, has limited NSC differentiation efficiency, and does not allow differentiated cell morphology observation [10].

The patterning of NSCs into MNPs follows that of the human neural tube development of the CNS. Reportedly, retinoic acid (RA) and sonic hedgehog (SHH) promote NSCs into posterior and ventral neural tube positions to produce MNPs. Thus, the addition of RA and SHH or their analogs is standard procedure for MNP patterning [13,14,15]. However, the MN differentiation efficiency remains limited (30–70%) in this or similar protocols.

To accelerate MN maturation to express late-stage markers, such as HB9 and ChAT, and induce neuronal electrophysiological function and NMJ formation, neurotrophic factors and Notch inhibitors, including bone-derived neurotrophic factor (BDNF), glial cell line-derived neurotrophic factor (GDNF), DAPT, and compound E are used to promote ChAT^+^ neuron formation with electrophysiological responsiveness within 35 days of differentiation [16,17].

To routinely produce MNs for MND studies, a stable supply of mature, ready-to-use MNs is critical. The cryopreservation of neuronal cells with high viability that retains their properties is difficult. Although there have been current studies that have reported the cryopreservation of iPSC-derived dopaminergic progenitors for Parkinson’s disease transplantation [18,19,20], only a few have achieved MN or MNP cryopreservation but without further functional analysis [21]. Moreover, most reports demonstrated that neuronal progenitors or stem cells are frozen within 10–20 days of differentiation instead of late-stage premature neurons (>30 days) [18,19,20], requiring additional maturation to recapitulate disease phenotypes for model applications.

In our previous report, we developed an SFEB-based NSC induction protocol (the BiSF protocol). The FGFb, activin inhibitor SB431542 (SB), and Wnt agonist BIO are added to rapidly convert PSCs into NSCs, potentially achieving various neural lineages with only 2 days of induction [22,23]. This protocol overcomes the major challenges of SFEB neuronal induction. Here, we established a CHSF-MN protocol based on the BiSF protocol to effectively convert PSC lines (including hESCs and iPSCs) into MNs. These near-mature-stage MNs can be cryopreserved and thawed with neurite extensions, promote high MN marker expression, and exhibit classical electrophysiological properties and NMJ formation ability. After applying the thawed CHSF-MNs for disease modeling, we successfully discovered ALS-specific pathogenic protein accumulation and redistribution in thawed SOD1^G85R^ and sporadic ALS MNs.

Herein, we developed a rapid, robust, and reproducible protocol to generate MNs from several PSC lines. Moreover, we could cryopreserve these near-mature-stage MNs to recapitulate ALS-specific phenotypes for off-the-shelf disease studies or drug screening without requiring additional maturation processes or prolonged culture.

## 2. Results

### 2.1. Establishment of the CHSF-MN Protocol to Differentiate iPSCs into MNPs and MNs

For MND modeling, we first generated an ALS iPSC line with the *SOD1* G256C point mutation (SOD1^G85R^). In our previous study, we employed CRISPR/Cas9 gene editing to correct the *SOD1* G256C mutant site to the normal nucleic acid, producing healthy SOD1 protein (SOD1^G85G^ iPSCs) [24]. In another previous study, we established an SFEB-based robust neuronal differentiation protocol known as the BiSF method to induce neuronal differentiation by adding FGFb, activin inhibitor SB, and Wnt ligand BIO (Figure 1a) [22,23]. Using the BiSF method, we converted 90% of SOD1^G85G^ iPSCs into neuroepithelial (NE) cells expressing Sox1 and N-cadherin (Figure 1c). According to the traditional MN patterning protocol, SHH analog SAG and RA were used to induce NE cells into the ventral part of the posterior neural tube (Figure 1a), which is abundant in MNs and interneurons in CNS development. However, this MN induction protocol did not exhibit consistent results among the experiments. In most batches, iPSCs could be converted into high-purity Sox1^+^ (94.43% ± 5.95%) and Oligo2^+^ (87.13% ± 2.08%) neural progenitors at day 15 of differentiation (Figure 1b,c). However, although these progenitor cells could become NF^+^ neurons (84.57% ± 4.92%; Figure 1g), they inconsistently differentiated into MNs, presenting only 40.49% ± 19.05% of HB9^+^/diamidino-2-phenylindole (DAPI) (with large variance from 22.46% to 60.42%), 55.74% ± 31.21% of Islet1^+^/DAPI, and 6.26% ± 6.36% ChAT^+^/βIIITUB expression at day 25 (Figure 1e–g), demonstrating that Oligo2 is not an MNP-specific marker. Furthermore, high Nkx2.2 levels were expressed at days 15 and 25 of differentiation (87.45% ± 4.81% and 52.44% ± 4.12%, respectively; Figure 1d,h). Nkx2.2 is a ventral spinal cord marker expressed in the common pool of Oligo2^+^ neural progenitors that finally segregate into HB9^+^Nkx2.2^−^ MNs and HB9^-^Nkx2.2^+^ interneurons [25,26]. Our Nkx2.2 expression results suggested the possibility that some Nkx2.2^+^ progenitors at day 15 may finally differentiate into interneurons instead of MNs at day 25. Moreover, BIO was found to induce a high cell death rate during the neural induction of some iPSC lines.

To overcome the two challenges of the BiSF protocol, including BIO-induced cell toxicity and inefficient HB9^+^ MN differentiation, we first replaced BIO with the dosage-dependent GSK3β inhibitor CHIR99021 (CHIR) and successfully reduced the toxicity effect of BIO on iPSCs (for data not shown). In our previous study, we found that BiSF neural induction for 2 days engendered anterior CNS fate, whereas BiSF treatment for 4 days inhibited the expression of the forebrain marker Pax6 [22]. Other studies also suggested that long-term Wnt activation may promote posterior neural differentiation [25,27]. Moreover, Wnt signaling is a key regulator of the distinct fate of Oligo2^+^Nkx2.2^+^ neural progenitors into MNs or interneurons [25,28]. Thus, CHIR was added to achieve better posterior patterning and MN fate determination and avoid the interneuron lineage. In CNS development, several morphogens are responsible for the anterior–posterior (A–P) and dorsal–ventral (D–V) determination, including RA (A–P), SHH (D–V), bone morphogenetic proteins (BMPs) (D–V), and Wnt (both A–P and D–V). CHIR activity not only promotes posterior MN fate determination but also dorsalization during the D–V development of the neural tube [13]. Thus, to avoid the complex interactions of several signaling pathways on D–V balance, SMAD inhibitors LDN193189 (LDN) and SB were applied to block BMPs and develop a modified protocol (CHSF-MN; Figure 1i). Using this protocol, we successfully obtained 95.81% ± 2.94% Sox1^+^/DAPI NSCs, most with N-cadherin co-expression (Figure 1k) at day 15 and 98.41% ± 0.21% NF^+^/DAPI neurons at day 25 of differentiation (Figure 1o), demonstrating that replacing BIO with CHIR does not decrease the neural induction efficiency. The high-level expression of Oligo2^+^/DAPI (95.11% ± 1.08%; Figure 1j) and low-level expression of interneuron-related marker Nkx2.2^+^/DAPI (7.05% ± 2.13%; Figure 1l) on day 15 suggested the emergence of a high-purity MNP population. Following an additional 10 days of maturation (a total of 25 days of differentiation), patterned cells expressed 96.48% ± 0.66% Islet1^+^/DAPI, 94.51% ± 1.99% HB9^+^/DAPI, 91% ± 2.65% Islet1^+^HB9^+^/DAPI, and 91.71% ± 2.36% ChAT^+^/βIIITUB (Figure 1m–o), suggesting reproducible high-efficiency MN differentiation. Previous studies have reported that the MN development-related markers, Oligo2 and Islet1, might be expressed in the MNPs (Oligo2) and premature MNs (Islet1) sequentially [25,27,29]. Thus, on day 15 of differentiation, most cells expressed Oligo2 but only a few expressed Islet1 (Figure 1j). Islet1 expression emerged on day 25 and was co-expressed with another premature MN-specific protein, HB9 (Figure 1n), suggesting that Oligo2^+^ MNPs on day 15 were successfully converted to premature MNs on day 25. The low Nkx2.2 expression on day 15 (7.05% ± 2.13%; Figure 1l) and undetectable Nkx2.2 expression (Figure 1p) on day 25 demonstrated the inhibition of interneuron marker Nkx2.2 in the CHSF-MN protocol to promote HB9^+^ MN differentiation. Marker expressions on day 15 (Figure 1q) and day 25 (Figure 1r,s) were calculated and are presented in Figure 1q–s.

### 2.2. CHSF-MN Protocol Efficiently Converts hESC and iPSC Lines into High-Purity MNs

To evaluate the applicability of the CHSF-MN protocol on other PSC lines, two hESC lines, H9 (from WiCell; Figure 2a–c) [30] and TW1 (established in Taiwan; Figure 2d–f) [31]; and 5 ALS-related iPSC lines, SOD1^G85R^ (Figure 2g–i) [24], SOD1^G85G^ (Figure 1) [24], SOD1^D90D^ (WiCell; Figure 2j–l) [6], SOD1^D90A^ (WiCell; Figure 2m–o) [6], and sporadic ALS (Figure 2p–r) [24] iPSC lines were directed toward MNs using this protocol. The differentiated cells expressed 95.43% ± 3.01% to 98.19% ± 1.05% Sox1 and 92.93% ± 1.36 to 98.13% ± 0.96% Oligo2 on day 15 with rosette-like structures in all hESC and iPSC lines (Figure 2s). On day 25, 96.92% ± 1.03% to 99.23% ± 0.19% of the differentiated cells expressed NF and 91.34% ± 4.26% to 93.54% ± 1.34% expressed HB9 in all PSC lines (Figure 2t). These data suggested that the CHSF-MN protocol efficiently generated MNs from various PSCs, including hESCs and iPSCs, as a reproducible and stable method to produce MNs.

### 2.3. Cryopreserved MNs Formed SFEBs, Exhibited Neurite Morphology, Expressed MN-Specific Markers, and Demonstrated Electrophysiological Properties

We aimed to cryopreserve late-stage premature MNs at 32 days of differentiation, following the expression of HB9, NF, and ChAT. We first screened commercial cell cryopreservation solutions, including Stem-CellBanker (Amsbio, Cambridge, MA, USA), PSC Cryopreservation Kit (ThermoFisher Scientific, Waltham, MA, USA), CryoStor cell cryopreservation media (including CS10 and CS5; Sigma-Aldrich, Saint Louis, MO, USA), and DMSO-based serum or serum-free solutions. After thawing, the embryoid body (EB) morphology (including the sphere size, surface properties of EBs, and single-cell number in the culture medium following 24 h of thawing) was examined and it was determined that Stem-CellBanker provided the best survival after thawing (data not shown). The cryopreserved MN spheres showed 88.91% cell viability. Therefore, we selected Stem-CellBanker as the cryopreservation media for MNs. To retain the specific marker expression of frozen MNs, we dissociated MN spheres into different-sized aggregates, including large (>500 μm), medium (300–500 μm), and small (200–300 μm) sphere sizes, before freezing, with or without adding the ROCK inhibitor (ROCKi) in the first 24 h after thawing. We found that large sphere sizes (>500 μm) exhibited the highest number of floating single cells in the medium, and the small spheres (200–300 μm) exhibited the best EB properties (Figure 3a). To assess neurite formation and MN-specific marker expression, we plated the 1-day thawed spheres on Matrigel-coated dishes. After 3 days, numerous neurite-like structures extended from the attached spheres (200–300 μm sphere size; Figure 3b). Furthermore, sphere size and ROCKi treatment altered the expression of the MN marker Islet1 (Figure 3c–h). Small sphere size alongside thawing ROCKi treatment demonstrated the best Islet1 expression rate (89.42% ± 3.97%) with neurofilament expression neurites, whereas large spheres without ROCKi demonstrated only 43.56% ± 5.73% Islet expression (Figure 3i).

To determine the maturation status and electrophysiological function of the thawed MNs, cells were subjected to ICC, single-cell patch-clamp, and calcium imaging. After 1 day of floating and 3 days of adherent culture (36 days of total differentiation), MNs from four PSC lines, H9 hESC, SOD1^G85G^, SOD1^G85R^, and sporadic ALS iPSC, expressed the mature MN protein ChAT (89.31% ± 2.83% to 90.39% ± 0.97% in all PSC line-derived MNs; Appendix A). In the single-cell patch-clamp analysis (Figure 4a), cryopreserved MNs responded to −80 to 100 mV voltage stimulation alongside Na+-associated inward and K+-associated outward currents (Figure 4b). Neuronal membrane properties were analyzed via depolarizing current pulses (200 pA) and firing action potentials (Figure 4c). MNs also responded to the glutamate treatment (Figure 4d), confirming the classical response to their specific neurotransmitter. These MNs exhibited strong calcium flux with potassium chloride and glutamate stimulations (Figure 4e), showing that most MN cells possess normal glutamate neuron functional properties.

### 2.4. Establishment of NMJ-like Structures via Thawed MN-Myoblast Coculture

The NMJ is the major structure that delivers signals from MNs to the muscles for effective myocontractions. In ALS and SMA, NMJ defects are the earliest cytopathy observed in human and transgenic animal tissues [32,33]. The NMJ formation ability of MNs represents an important property of muscle linkage and NMJ cytopathy recapitulation. Thus, thawed MNs derived from ALS (SOD1^G85R^) and mutant site-corrected healthy (SOD1^G85G^) iPSC lines were cocultured with mice myoblast cell lines (Figure 5a). After 6 days of coculture, both SOD1^G85R^ and SOD1^G85G^ MNs formed NMJ-like structures with overlapping neurofilaments (NF), myotubes(myosin heavy chain [MHC]), and acetylcholine receptor clumps (αBTX) with both C2C12 [34] and Sol8 [35] myoblasts (Figure 5b–e). To identify ALS NMJ cytopathies, the intensity of acetylcholine receptor (αBTX+ area) was normalized with myotube (MHC+ area), and the expression of acetylcholine receptor did not exhibit significant differences between SOD1^G85G^ and SOD1^G85R^ NMJ cocultures (data not shown). In the experiment involving myotube contraction, we found that C2C12 myoblasts promoted spontaneous contractions in the SOD1^G85G^ MN coculture (Appendix A), very few in the SOD1^G85R^ MN coculture (Appendix A), and none in C2C12 alone (Appendix A). To determine whether myotube contractions were induced by functional NMJ, an NMJ inhibitor was added to the MN-C2C12 coculture. Myotube contractions decreased in both SOD1^G85G^ (Appendix A) and SOD1^G85R^ (Appendix A) MN cocultures following NMJ inhibitor treatment, suggesting that thawed MNs formed functional NMJs and potentially induced the NMJ cytopathy of ALS.

### 2.5. Thawed ALS MNs Expressed Classical Pathogenic Protein Aggregates and Redistribution Cytopathies

To further examine whether thawed MNs could be applied to ALS modeling, we differentiated ALS iPSC lines, including SOD1^G85R^ and sporadic ALS, isogenic healthy iPSC SOD1^G85G^, and H9 control hESCs, into MNs using the CHSF-MN protocol. Following cryopreservation and thawing, an accumulation of misfolded SOD1 aggregates were observed in the soma of SOD1^G85R^ and sporadic ALS MNs (Figure 6e,g). The redistribution of the nucleus protein TAR DNA-binding protein 43 (TDP-43) in the cytosol of neurons was previously associated with frontotemporal dementia and TDP-43 mutant ALS [36,37]. Recently, the TDP-43 redistribution pathology was also observed in the cell and animal models and in patient samples of SOD1 mutant and sporadic ALS [8,38]. Thus, we identified the TDP-43 location in thawed MNs and found the SOD1^G85R^ MNs exhibited TDP-43 relocalization in the cytosol (Figure 6c,d) as an early sign of TDP-43 accumulation, serving as another classical ALS amyloid-like cytopathy. Compared with ALS MNs, SOD1^G85G^ and H9 MNs did not exhibit SOD1 aggregates, and sporadic ALS did not show TDP-43 redistributions (Figure 6a,b,f), demonstrating the specificity of disease phenotypes in ALS MNs.

## 3. Discussion

We developed a robust protocol to convert human PSC lines into MNPs and MNs within 25 days of differentiation. Using this protocol, **CH**IR, **S**B, and **F**GFb (CHSF) were applied to induce NSC differentiation, followed by MN patterning with SHH, RA, Wnt activator CHIR, and SMAD inhibitors LDN and SB. In the seven tested PSC lines, the same protocol was used alongside the same differentiation conditions, demonstrating the formation of >91% HB9^+^ MNs in all the PSC lines. This protocol overcame the previous challenges of the SFEB-based methods, such as long-term NSC induction and differentiation efficiency variation within individual batches or PSC lines. Therefore, this protocol is an easily accessible, reproducible method of generating MNs without tweaking adherent conditions, cell confluence, or passage timing in monolayer differentiation. Furthermore, it is suitable for good manufacturing process development.

Wnt activation plays a dominant role in the posterior decision and motor neuron–interneuron determination during neural tube development. In our previous BiSF neural induction protocol, the Wnt analog promoted robust NSC conversion. Moreover, BIO addition for 4 days eliminated Pax6 expression, whereas 2 days of BIO treatment retained forebrain markers, including Pax6, BF-1, and Forse-1, suggesting a role of long-term Wnt activation in the posterior fate decision during PSC differentiation [22]. Du et al. also demonstrated that a Wnt analog not only contributed to posterior patterning but also induced Oligo2^+^Nkx2.2^−^ progenitors to differentiate into MNs during PSC differentiation [25]. Therefore, Wnt activation during neural induction and MNP patterning effectively induced MN fate. However, Wnt signaling is also involved in D–V determination. To avoid the interference of other signals in the D–V decision, BMP and activin pathways were blocked using their respective inhibitors, thereby controlling the SHH and Wnt balance and differentiating cells into a ventral fate.

In Figure 6, sporadic ALS iPSC line-derived MNs were found to express misfolded SOD1 aggregates, which has hardly been mentioned in previous reports. According to a recent study, misfolded SOD1 aggregates were identified in the sporadic ALS human tissues, indicating a SOD1 pathology in sporadic ALS [39]. However, very few reports have focused on sporadic ALS iPSC-derived MNs to recapitulate or discover sporadic ALS-specific cytopathies. Thus, the role of SOD1 protein-related phenotypes in sporadic ALS remains largely unknown. In 2018, Fujimori et al. established 32 iPSC lines from patients with sporadic ALS and found that certain sporadic ALS iPSC line-derived MNs showed SOD1 aggregates. However, most sporadic ALS line-derived MNs were rarely associated with SOD1-related phenotypes [8]. Their report suggested the existence of SOD1-related cytopathies in sporadic ALS MNs. To evaluate the proportion of SOD1-related phenotypes in sporadic ALS, more sporadic ALS iPSC lines should be included for disease modeling.

The storage of MNs at a near-mature stage to enable immediate application to disease modeling after thawing without requiring further complex patterning or maturation processes is extremely important for large-scale, reproducible drug screening and establishing complex MND models, such as NMJ, organoid, and microfluid organ chips. However, few studies have reported protocols for cryopreserving near-mature PSC-derived neurons with fully identified neuronal properties after thawing. Most studies have demonstrated NSC or progenitor-stage cell storage for ~15 days of differentiation, which still require another 2–3 weeks for neural patterning or maturation [18,19]. Here, we demonstrate an efficient protocol to cryopreserve MN EBs for 32 days using Stem-CellBanker with a sphere size of <300 µM and ROCKi addition. This cryopreservation protocol imparts MNs with a high survival rate, MN-specific marker expression, neurite extensions, and classical electrophysiological properties capable of NMJ formation after thawing. For ALS modeling, we explored myocontraction differences between SOD1^G85G^ control and SOD1^G85R^ ALS MN on the C2C12 coculture NMJ model. Moreover, protein abnormalities, including misfolded SOD1 accumulation and TDP-43 redistribution in thawed SOD1^G85R^ ALS MNs and SOD1 aggregates in sporadic ALS MNs were identified. Thus, thawed ALS MNs expressed patient tissue-like cytopathies.

## 4. Materials and Methods

### 4.1. Pluripotent Stem Cell Culture and Expansion

The H9 hESC, SOD1^D90A^, and SOD1^D90D^ iPSC lines were purchased from WiCell (Madison, WI, USA) [6]. TW1 hESCs were kindly provided by Lee Women’s Hospital (Taichung, Taiwan) [31]. SOD1^G85R^ and sporadic ALS iPSC lines were generated from patients’ peripheral blood mononuclear cells (PBMCs) with the approval of the Ethical Institutional Review Board at Hualien Tzu Chi Hospital (IRB105-131-A), and SOD1^G85G^ iPSCs were generated from SOD1^G85R^ iPSCs [24]. Informed consent was obtained from all the patients. The iPSC properties were previously identified. Briefly, PBMCs were cultured in StemPro-34 SFM (ThermoFisher Scientific) and supplemented with SCF, FLT-3, IL-3, and IL-6 (Peprotech, Cranbury, NJ, USA). Reprogramming was performed using CytoTune iPS 2.0 Sendai Reprogramming Kit (ThermoFisher Scientific) as per the manufacturer’s instructions [40]. All hESC and iPSC lines were transferred onto 1% Geltrex (ThermoFisher Scientific)-coated culture dishes and expanded in the TeSR-E8 media (StemCell Technologies, Vancouver, BC, Canada) or DuoESy pluripotent stem cell culture media (Without Geltrex coating, DuoGenic Stem Cells, Taichung, Taiwan). The hESCs and iPSCs were passaged for 3–5 days using Accutase (Merck-Millipore, Billerica, MA, USA) and then reseeded at 1:5–1:10 ratios. Culture media were refreshed daily.

### 4.2. MNP Differentiation and MN Maturation

When the hESCs and iPSCs reached 80% confluence in a culture dish, the cells were treated with 1 mg/mL Dispase II (Sigma-Aldrich) for 5–15 min until the periphery of the colonies started to round up. The cells were washed with DMEM-F12 twice and then lifted by scraping. The clumps were dissociated into ~200 μm clusters by pipetting and transferred into ultralow attachment dishes in the TeSR-E6 media (StemCell Technologies) for 24 h to form SFEBs. For successful EB formation, 5 μM ROCKi Y27632 (Cayman Chemical, Ann Arbor, MI, USA) was added during the first 4 h to avoid PSC blebbing. The EBs were subjected to CHSF neural induction. The neural induction media comprised DMEM-F12 with 1% N2 supplement, 1 mM NEAAs, and 2 mM glutamate (all from ThermoFisher Scientific). CHSF neural induction factors, including 3 μM CHIR99021 (Cayman Chemical), 10 μM SB431542 (Sigma-Aldrich), and 10 ng/mL recombinant human FGF-2 (rh-FGF2; R&D Systems, Minneapolis, MN, USA), were added on days 1–4 for 3 days without changing the media. On day 4, the neural induction media were changed to neurobasal media (ThermoFisher Scientific) with 1% N2 supplement, 1 mM NEAAs, 2 mM glutamate, and MNP patterning factors (3 μM CHIR99021, 2 μM SB431542, 200 nM LDN193189 (Cayman Chemical), 0.5 μM SAG (Cayman Chemical), and 0.5 μM RA (Sigma-Aldrich)) until day 25 of differentiation. On day 15, 2% B27 supplement (ThermoFisher Scientific) was also added. After day 25, morphogens were withdrawn from the culture medium. The EBs were mechanically pipetted into 200–300 µm per 2–3 days and cultured for at least another 4–8 weeks. After the end of the neural induction step on day 4, the culture media was changed every other day.

### 4.3. MN Cryopreservation and Thawing

On day 32, EBs were dissociated into 200–300 µM clumps using mechanical pipetting and centrifuged at 1000 rpm for 2 min. We removed the supernatant, gently tapped the tube to dissociate the pellet, suspended ~100 EBs in 1 mL STEM-CELLBANKER Stem Cell Freezing Media (Amsbio) supplemented with ROCKi and 1% RevitaCell (ThermoFisher Scientific), and transferred it to 1-mL cryotubes. The cryotubes were placed in a 4 °C precooled Cryo 1 °C freezing container (Nelgene) filled with fresh isopropanol. The freezing container was placed in a −80 °C refrigerator for 24 h and transferred into a liquid nitrogen tank for long-term storage. For thawing, the cryotube was transferred into a 37 °C water bath until a small amount of unmelted ice remained. EBs were quickly transferred into a 15-mL centrifuge tube, immediately supplemented with 2 mL DMEM-F12, and centrifuged at 1000 rpm for 2 min. The supernatant was discarded, the tube was tapped to loosen the pellet, and EBs were transferred to an ultralow culture dish with neurobasal media supplemented with 1% N2, 1 mM NEAAs, 2 mM glutamate, and 2% B27. In the first 24 h of thawing, 1% RevitaCell supplement was also added. For adherent culture or further experiments, EBs were dissociated using mechanical force or Accutase and seeded on 1% Geltrex-coated cell culture dishes supplemented with 0.2 μM compound E (Cayman Chemical), BDNF (Peprotech), GDNF (Peprotech), and dbcAMP (Sigma-Aldrich) for 3 days. In the first 4 h of cell adhesion, 1% RevitaCell supplement (ThermoFisher Scientific) was added to the cells.

### 4.4. ICC

The cells were seeded in 24-well plates and fixed with 4% paraformaldehyde (Sigma-Aldrich) at room temperature and then washed twice with phosphate-buffered saline. Cells were permeabilized using 0.1% Triton-100 for 15 min at 4 °C and blocked with 5% horse serum. Primary antibodies were incubated with the cells at 4 °C for 16 h, washed twice, and then conjugated with secondary antibodies. The primary antibodies used in this study included those against N-cadherin (1:500) and Sox-1 (1:200) (Santa Cruz Biotechnology); Nestin (1:500) and β-III-tubulin (1:500) (Covance, Princeton, NJ, USA); NF (1:500), phosphorylated neurofilament H & M (1:500), and choline acetyltransferase (1:100) (Sigma-Aldrich); Nkx2.2 (1:50), myosin heavy chain (1:50), and HB9/MNX1 (1:50) (Developmental Studies Hybridoma Bank, DSHB, Iowa City, IA, USA); Islet1 (1:500; and Neuromics, Edina, MN, USA), Oligo2 (1:500; Novus Biologicals, Centennial, CO, USA), TDP-43 (1:500; Proteintech, Tokyo, Japan), and misfolded SOD1 (1:250; Medimabs, Montréal, QC, Canada). The acetylcholine receptor was labeled with alpha-bungarotoxin (α-BTX). Cell nuclei were stained with DAPI. Fluorescence images were captured with an upright microscope (Imager Z1, Zeiss, Oberkochen, Germany). For the quantification of specific marker expression, images from three independent experiments were counted using Image J software (National Institutes of Health, Bethesda, MD, USA). Sox-1, Nkx2.2, HB9, Islet1, NF, and Oligo2 were normalized and presented as the ratio (%) of DAPI^+^ cells. DAPI^+^ cell nucleus was identified using 400× magnification. Each identifiable DAPI^+^ single nucleus that did not overlap with other nuclei was included for specific marker counting. Choline acetyltransferase was normalized as the ratio (%) of β-III-tubulin^+^ neurites.

### 4.5. Whole-Cell Patch-Clamp Recording

Two solutions were used in this protocol: artificial cerebral spinal fluid (aCSF) comprising 127 mM NaCl, 3 mM KCl, 26 mM NaHCO_3_, 1.25 mM NaH_2_PO_4_, 2 mM CaCl_2_, 1 mM MgSO_4_, and 10 mM D-glucose (pH 7.45). The pipette solution constituted 140 mM K gluconate, 10 mM NaCl, 0.5 mM EGTA, 10 mM HEPES, 3 mM ATP-Mg, and 0.4 mM GTP (pH 7.3, adjusted with KOH). Electrodes were pulled from a 1.5-mm outer diameter and 1.0-mm inner diameter capillary glass (World Precision Instruments PG52151-4) using Sutter P-97 puller (Sutter Instrument, Novato, CA, USA), and the tips were slightly fire-polished using MF-830 microforge (Narishige, Tokyo, Japan). After filling with pipette solution, electrode impedance in the aCSF solution was 6–15 megaohms (MΩ). The series resistance was not compensated. The junction potential was zeroed. The PSC-derived MNs on coverslips were transferred to the recording chamber, which was continuously perfused with aCSF. The MN responses were recorded using Axoclamp 200B (Axon Instruments, Union City, CA, USA). Voltage clamp mode was used to examine ion channel or receptor-mediated currents and intrinsic membrane properties. Current-clamp mode was used to investigate the firing properties of MNs. Cells were defined as neurons only if they exhibited fast-inactivating inward currents. Three recording protocols were used to elicit cell responses. (1) The membrane potential was maintained at −60 mV. Then, the membrane potential was increased from −80 to +40 mV in 10 mV increments for 400 ms to determine the presence of sodium and potassium currents. (2) In the current-clamp mode, current steps from −60 to +120 pA in 20 pA increments were applied to the cells to examine their ability to generate APs. (3) To evaluate the response of neurons to glutamate, the cells were maintained at −60 mV for 2–10 s in the presence of 1 μM glutamate to record spontaneous and receptor-mediated postsynaptic currents. Data were recorded and analyzed using Digidata 1322A and pClamp 9.0 (Axon Instruments, Inc.).

### 4.6. Calcium Imaging

Cells were seeded on Geltrex-coated 10-mm cover slides and cultured in a physiological buffer with 1 μM Fluo-4 at 37 °C for 40 min. Subsequently, they were incubated in physiological buffer (including 30 mM NaCl, 5 mM KCl, 2 mM CaCl_2_, 2 mM MgCl_2_, 10 mM glucose, and 10 mM HEPES) for 20 min. The cells were moved to a calcium imaging chamber, washed with physiological buffer, and treated with 60 mM of KCl for 60 s. After 5-min washing with physiological buffer, the cells were treated with 1 mM L-glutamate for 60 s. Images were captured using the ECLIPSE Ti2-E microscope (Nikon, Tokyo, Japan) and analyzed using NIS-Elements AR (Nikon).

### 4.7. Mouse Myoblasts and MN Coculture

The mouse myoblast cell line sol8 was cultured in a proliferation medium comprising DMEM with sodium pyruvate supplemented with 20% fetal bovine serum (FBS), 1 mM NEAAs, and 2 mM glutamate (all obtained from ThermoFisher Scientific). Cells were subcultured with 0.25% trypsin/EDTA (ThermoFisher Scientific) when they reached >50% confluence and then reseeded with 3 × 10^5^ cells per 10 cm cell culture dish. For myotube differentiation, sol8 cells were seeded with 5 × 10^4^ cells in a 24-well plate, and the FBS in the culture media was replaced with 2% horse serum (ThermoFisher Scientific) for 6 days before the MN coculture. The other mouse myoblast cell line C2C12 was cultured in media comprising DMEM supplemented with 10% FBS, 1 mM NEAAs, and 2 mM glutamate. Cells were subcultured with 0.25% trypsin/EDTA when they reached >50% confluence and then reseeded with 4 × 10^5^ cells per 10 cm cell culture dish. For myotube differentiation, C2C12 cells were seeded with 1.5 × 10^5^ cells in a 24-well plate, and the FBS in the culture media was replaced with 2% horse serum for 6 days before MN coculture. For MN culture, the media for differentiated myotubes were replaced with neurobasal media supplemented with 1% N2, 1 mM NEAAs, 2 mM glutamate, and 2% B27. The thawed MN EBs were added to myotubes for 6 days before NMJ analysis. Lastly, 200 μM curare (Sigma-Aldrich) was added into MN-myotube coculture to inhibit NMJ formation.

### 4.8. Statistical Analysis

Data were collected and analyzed from no less than three independent experiments and are shown as means ± SD. Student’s *t*-test or one-way ANOVA analysis of variance with Tukey’s post hoc test was used to compare the groups. *p*-values of <0.05 were considered statistically significant.

## 5. Conclusions

We established a reproducible protocol to efficiently convert PSCs into MNs. With the development of this novel near-mature MN cryopreservation method, we provide patient-derived MNs suitable for ALS modeling without the need for further processing. These ready-to-use MNs can be applied in large-scale, high-throughput drug screening and to establish complex MND models, such as NMJ, organoid, and microfluid organ chips, for precision medicine development.

## Figures and Tables

**Figure 1 ijms-23-13462-f001:**
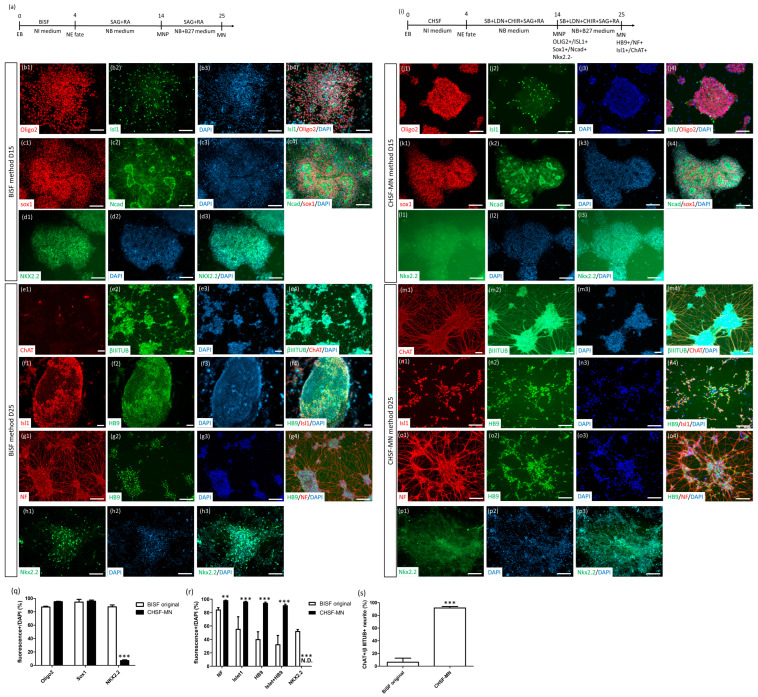
CHSF-MN protocol improved the MN generation ability and stability of the BiSF protocol. (**a**) Procedure of BiSF-based MN differentiation. (**b**–**h**) Identification of SOD1^G85G^ iPSC line-derived MNPs and MNs via the BiSF protocol using neural stem cell markers (Sox1 and N-cadherin), MNP markers (Oligo2 and Islet1), and an interneuron marker (Nkx2.2) on day 15 and MN (HB9, Islet-1, and ChAT) and neuronal (NF and β-III-tubulin) markers on day 25 of differentiation. (**i**) Procedure of the CHSF-MN protocol. (**j**–**p**) Identification of MNPs and MNs obtained via the CHSF-MN protocol using neural stem cell markers (Sox1 and N-cadherin), MNP markers (Oligo2 and Islet1), and an interneuron marker (Nkx2.2) on day 15 and MN (HB9, Islet1, and ChAT) and neuronal (NF and β-III-tubulin) markers on day 25 of differentiation. (**q**–**s**) Calculation of the related marker expression ratio of DAPI^+^ or β-III-tubulin^+^ (βIIITUB) cells on days 15 (**q**) and 25 (**r**,**s**) of differentiation. Number 1–3 presenting the separated images and number 4 presenting merged images in (**b**–**h**) and (**j**–**p**). ** *p* < 0.01 and *** *p* < 0.001. Scale bar, 100 μm.

**Figure 2 ijms-23-13462-f002:**
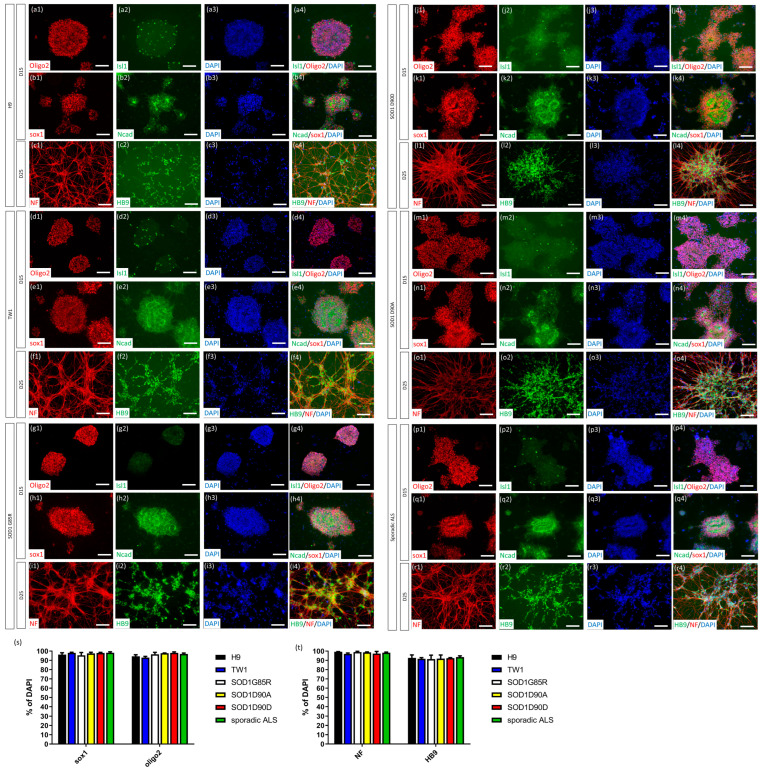
Robust generation of MNs from PSCs using the CHSF-MN protocol. Identification of hESC lines, including H9 (**a**–**c**), TW1 (**d**–**f**), and iPSC lines, such as SOD1^G85R^ (**g**–**i**), SOD1^D90D^ (**j**–**l**), SOD1^D90A^ (**m**–**o**), and sporadic ALS (**p**–**r**)-derived MNPs and MNs, via the CHSF-MN protocol using neural stem cell (Sox1 and N-cadherin) and MNP (Oligo2 and Islet1) markers on day 15 and MN (HB9) and neuronal (NF) markers on day 25 of differentiation. (**s**,**t**) Calculation of the marker expression ratio of DAPI^+^ cells on days 15 (**s**) and 25 (**t**) of differentiation. Number 1−3 presenting the separated images and number 4 presenting merged images in (a−r). Scale bar, 100 μm.

**Figure 3 ijms-23-13462-f003:**
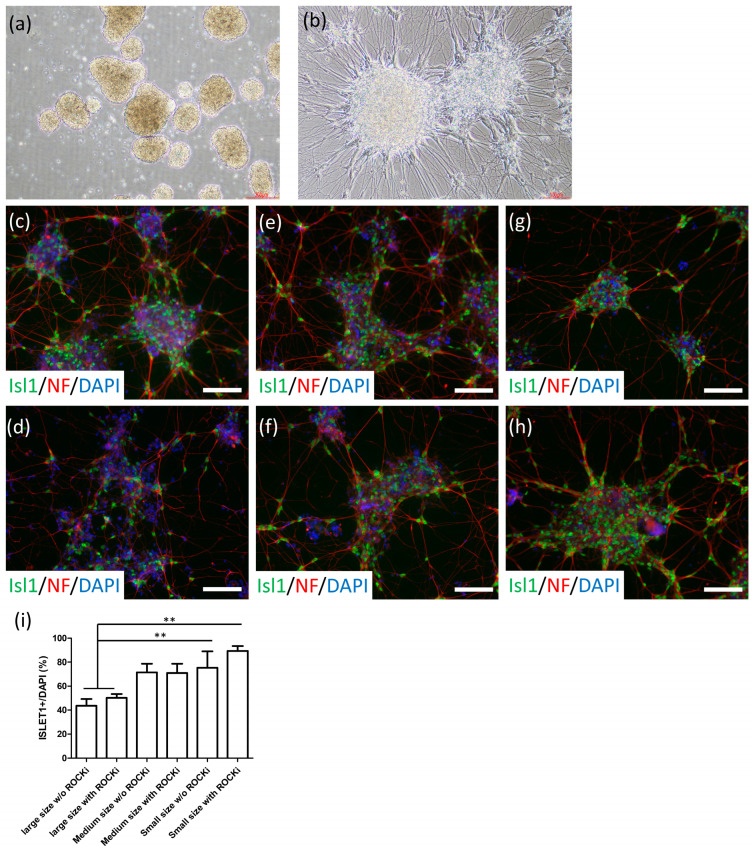
Cryopreservation and thawing of MNs on day 32. (**a**) Morphology of MN spheres after 24 h of thawing. (**b**) Morphology of adherent MNs after 4 days of thawing. (**c**–**h**) MN marker Islet1 (Green) and neuronal marker NF (Red) expression in DAPI^+^ cells (Blue) after 4 days of thawing with >500 µM EB without ROCK inhibitor treatment (**c**), >500 μM EB with ROCK inhibitor (**d**), 300–500 μM EB without ROCK inhibitor treatment (**e**), 300–500 μM EB with ROCK inhibitor treatment (**f**), 200–300 μM EB without ROCK inhibitor treatment (**g**), and 200–300 μM EB with ROCK inhibitor treatment (**h**). (**i**) Calculation of the Islet-1 expression ratio of DAPI^+^ cells. ** *p* < 0.01. Scale bar, 100 μm.

**Figure 4 ijms-23-13462-f004:**
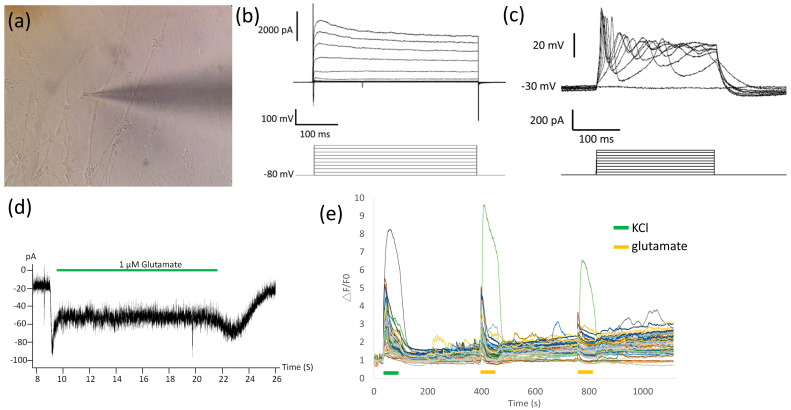
Classical functional electrophysiological properties of thawing MNs. (**a**) Image of patch-clamp cells and electrodes. (**b**) Voltage-gated ion current elicited via voltage stimulation. (**c**) Firing characteristics of MNs in response to 200 pA current injections. (**d**) Spontaneous and receptor-mediated postsynaptic currents via the glutamate treatment of MN. (**e**) Calcium flux changes of MNs via potassium chloride and glutamate stimulation.

**Figure 5 ijms-23-13462-f005:**
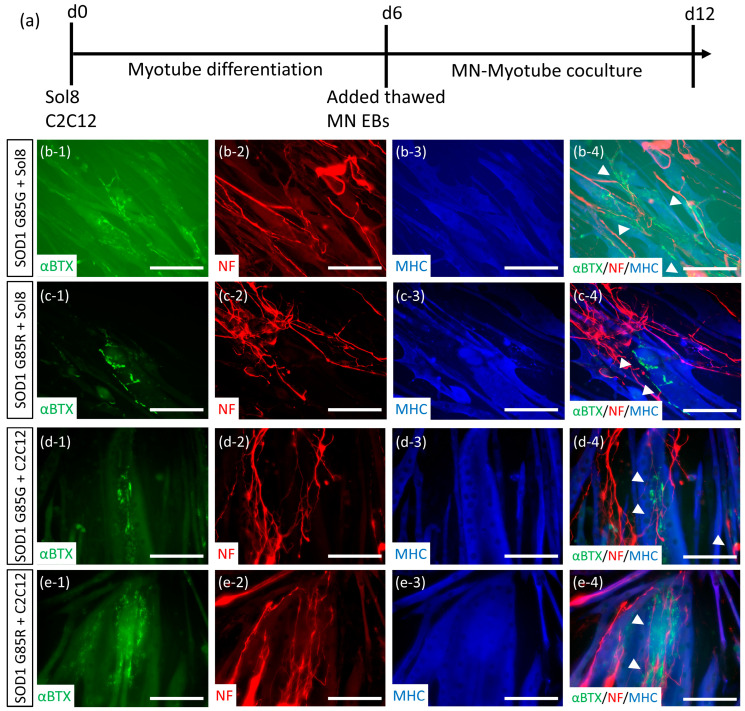
Thawed MNs formed a neuromuscular junction (NMJ)-like structure with mice myoblast cell lines. (**a**) The MN-myoblast coculture procedure. (**b**–**e**) Neuronal (NF), myotube (myosin heavy chain, MHC), and NMJ-specific marker (acetylcholine receptor, recognized with α-BTX) expression and colocalization of the SOD1^G85G^ MN-sol8 (**b**), SOD1^G85R^ MN-sol8 (**c**), SOD1^G85G^ MN-C2C12 (**d**), and SOD1^G85R^ MN-C2C12 coculture (**e**). Colocalizations of NF, MHC, and α-BTX are indicated by arrowheads. Number 1−3 presenting the separated images and number 4 presenting merged images in (b−e). Scale bar, 100 μm.

**Figure 6 ijms-23-13462-f006:**
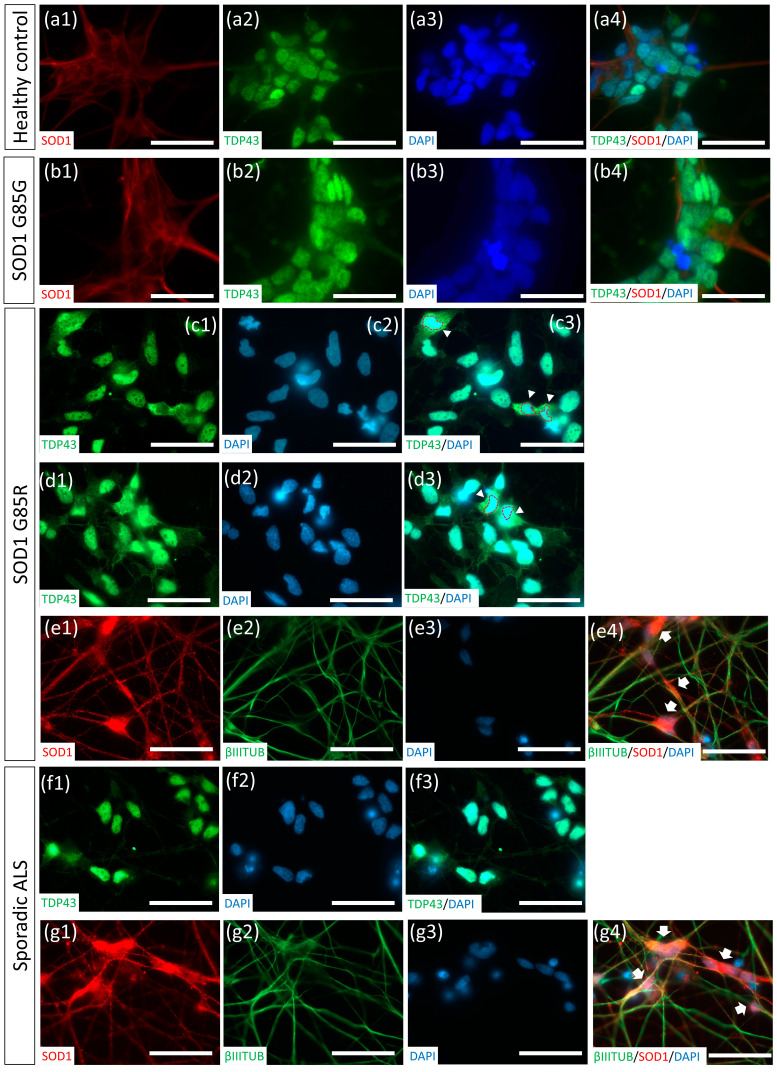
Misfolded SOD1 accumulation and cytosolic TDP-43 distribution cytopathies in SOD1^G85R^ and sporadic ALS MNs. (**a**–**g**) SOD1 and TDP-43 expressions in DAPI^+^ cells from healthy hESC-derived MNs (**a**), SOD1^G85G^ MNs (**b**), SOD1^G85R^ MNs (**c**–**e**), and sporadic ALS MNs (**f**–**g**). The cell nuclei are indicated by red circles and cytosolic TDP-43 is indicated by arrowheads in (**c**,**d**), respectively. Accumulated misfolded SOD1 is indicated by arrows in (**e**,**g**). Number 1−3 presenting the separated images and number 4 presenting merged images in (**a**−**g**). Scale bar, 100 μm.

## Data Availability

Data supporting the findings of this study are available in the figures and the results section. Further data are available from the corresponding author upon reasonable request.

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
