# Peer review of "Robust Generation of Ready-to-Use Cryopreserved Motor Neurons from Human Pluripotent Stem Cells for Disease Modeling"

_ijms, 2022, doi:10.3390/ijms232113462_

Round 1
Reviewer 1 Report
Major:
1) Fig. 1 and Fig. 2 demonstrating MN differentiation are not convincing. Individual markers should be presented separately (without superimposition), and the method of counting positive cells should be described. White field images should also be helpful. I cannot see any ChAT positive cells in Fig. 1, although the authors claim that most of the cells are positive, without providing any precise quantitation. Why no anti-ChaT marker was used in Fig. 2?
2) There is inconsistency in Fig. 5. SOD1-G95R + C2C12 coculture actually looks better in every parameter (marker) used than the isogenic (corrected) SOD1-G95G + C2C12 coculture. It should be the opposite according to the text and the logics.
3) In Fig. 6, TDP-43 mis-translocation should be explained, indicated, visualized in a better way. Possibly using a high amplification in comparative cell images.
4) Accumulation of misfolded SOD1 in sporadic ALS cell lines, Fig. 6, is not convincing. The signal is hardly associated with cells. And finding of SOD1 aggregation in sALS cell lines, especially in two randomly chosen different cell lines, is not trivial. This finding, if actual, should be properly discussed in the paper, with appropriate referencing and rationalization.
Minor:
1) l. 44. "are promising sources for human developmental and in vitro studies," Sources of what?
2) l. 60. "takes a longer time to produce NSCs, has limited NSC differentiation", Longer than what?
3) l. 65. "addition of RA and SHH or their agonists are standard procedures for MNP", Analogs? Agonists can be of receptors, not ligands.
4) l. 76. "derived dopaminergic progenitors for Parkinson’s disease transplantation." Any reference?
5) l. 120. "To overcome these two challenges, we first", Which challenges?
6) l. 131. "promotes posterior MN fate determination but also against the SHH ventralization", Against what? Acts against?
7) Fig. 6. Some arrow heads point to nowhere meaningful in the images. For instance, in panel c-1.
Author Response
Reply Letter
REVIEWER 1
Comments and Suggestions for Authors
Major:
1)Fig. 1 and Fig. 2 demonstrating MN differentiation are not convincing. Individual markers should be presented separately (without superimposition), and the method of counting positive cells should be described. White field images should also be helpful.
Reply: Thank you for your suggestions. We present the ICC results separately in the figure 1 and 2. The counting method is added in the material and method (line 401-405) and the numerator of expression ratio is added in the result (line 113-116, 137-143). If the figures integrated in the word or PDF file are not clear enough, please refer to the individual figure JPEG files for high resolution image.
I cannot see any ChAT positive cells in Fig. 1, although the authors claim that most of the cells are positive, without providing any precise quantitation.
Reply: The ChAT ICC in figure 1 is presented separately (figure 1e and 1m) and counted in figure 1s. We add the description at line 114 and 143. thank you for your suggestion.
Why no anti-ChaT marker was used in Fig. 2?
Reply: We are sorry that we didn’t do the ChAT ICC for all PSC line derived MNs before cryopreservation. To remedy this, we thawed the cryopreserved MN stocks and do the ChAT ICC, present as supplementary figure S1, and add the description in result at line 208-209.
2) There is inconsistency in Fig. 5. SOD1-G95R + C2C12 coculture actually looks better in every parameter (marker) used than the isogenic (corrected) SOD1-G95G + C2C12 coculture. It should be the opposite according to the text and the logics.
Reply: We are sorry to confuse the readers that the SOD1G85R+C2C12 NMJ markers looks better than the SOD1G85G. This may due to the picture we used. We choose the picture majorly for presenting the NMJ marker colocalization. Actually, we had observed the ICC coverslips, also tried to calculate the BTX expression/MHC area on both SOD1G85G and SOD1G85R C2C12 coculture but didn’t see significant NMJ marker expression differences between these two lines. To avoid this misunderstanding, we add the description at line 233-234 to claim there is no significant marker expression difference within these 2 lines in coculture model.
3) In Fig. 6, TDP-43 mis-translocation should be explained, indicated, visualized in a better way. Possibly using a high amplification in comparative cell images.
Reply: Thank you for your suggestion. We try to improve the TDP-43 presentation in figure 6 with separate images that may be helpful to observe the cell nucleus and TDP-43 individually (figure 6c and 6d). In the merged image (c3 and d3), we label the nucleus shape with red circles, and TDP-43 redistributions are indicated by arrowheads. The figure 6 JPEG file could provide high resolution pictures for clear image than the word or PDF. The description of TDP-43 redistribution is added at line 255-259 for better understanding.
4) Accumulation of misfolded SOD1 in sporadic ALS cell lines, Fig. 6, is not convincing. The signal is hardly associated with cells.
Reply: For better image, we thawed the SOD1G85R and sporadic ALS MN stocks and co-stain misfolded SOD1 with β-III-tubulin (figure 6e and 6g). The dot like SOD1 accumulations in the cell soma are indicated with arrows.
And finding of SOD1 aggregation in sALS cell lines, especially in two randomly chosen different cell lines, is not trivial. This finding, if actual, should be properly discussed in the paper, with appropriate referencing and rationalization.
Reply: In the figure 6, we used 1 healthy ES cell line, 1 SOD1G85R ALS iPSC line, 1 SOD1G85G corrected iPSC line and 1 sporadic ALS iPSC line. To present the SOD1 aggregates, we used 2 photos from 1 sporadic ALS line, not from 2 different sporadic ALS lines. I’m sorry to make reviewer confused. To avoid this misunderstanding, we modify the figure 6 and put only 1 image of SOD1 ICC for sporadic ALS on figure 6g.
Additionally, thank you for your reasonable suggestion. The SOD1 aggregates were rarely observed in sporadic ALS cases. We search the literature and find the misfolded SOD1 aggregates could be found in some patients’ tissues and sporadic ALS iPSC derived MNs. The discussion of SOD1 aggregates in sporadic ALS is added at line 296-308
Minor:
1) l. 44. "are promising sources for human developmental and in vitro studies," Sources of what?
Reply: This sentence is modified with “are promising sources of various cell types for human developmental and in vitro studies” at line 44-45
2) l. 60. "takes a longer time to produce NSCs, has limited NSC differentiation", Longer than what?
Reply: This sentence is modified with “takes a longer time than monolayer differentiation protocol to produce NSCs, has limited NSC differentiation” at line 60.
3) l. 65. "addition of RA and SHH or their agonists are standard procedures for MNP", Analogs? Agonists can be of receptors, not ligands.
Reply: agonists is changed with analogs at line 66.
4) l. 76. "derived dopaminergic progenitors for Parkinson’s disease transplantation." Any reference?
Reply: references 18-20 are added at line 77.
5) l. 120. "To overcome these two challenges, we first", Which challenges?
Reply: This sentence is modified with “To reduce the cell toxicity of BIO and improve the MN differentiation efficiency” at line 121-122.
6) l. 131. "promotes posterior MN fate determination but also against the SHH ventralization", Against what? Acts against?
Reply: This sentence is modified with “promotes posterior MN fate determination but also the dorsalization on the D–V decision” at line 133.
7) Fig. 6. Some arrow heads point to nowhere meaningful in the images. For instance, in panel c-1.
Reply: Sorry for the unclear label, we adjust the arrowhead and arrow labels of figure 6.

Reviewer 2 Report
The authors describe an improved method to convert pluripotent stem cells into motor neurons. Among the advantages of their protocol: production of mature MNs, high efficiency, high viability after thawing. Moreover, they show that this method can be used with different lines of pluripotent cells, including sporadic and familial ALS lines. Therefore, this work can be of interest for research groups that use iPSCs for neuromuscular disease modeling. I recommend publication after the following points are addressed:
- to demonstrate that the muscle contractions in the co-culture experiments are due to the establishment of functional NMJs, the authors should include experiments in which they stimulate the cells with glutamate, which should increase the contractions, or treat them with a specific inhibitor of the NMJ, such as curare, which should reduce the contractions.
- Images of the staining of the misfolded proteins in figure 6 should be improved by providing higher magnification figures.
- English language can be improved.
Author Response
Reply Letter
REVIEWER 2
Comments and Suggestions for Authors
The authors describe an improved method to convert pluripotent stem cells into motor neurons. Among the advantages of their protocol: production of mature MNs, high efficiency, high viability after thawing. Moreover, they show that this method can be used with different lines of pluripotent cells, including sporadic and familial ALS lines. Therefore, this work can be of interest for research groups that use iPSCs for neuromuscular disease modeling. I recommend publication after the following points are addressed:
- to demonstrate that the muscle contractions in the co-culture experiments are due to the establishment of functional NMJs, the authors should include experiments in which they stimulate the cells with glutamate, which should increase the contractions, or treat them with a specific inhibitor of the NMJ, such as curare, which should reduce the contractions.
Reply: We did the curare experiments but not included the results in the original manuscript. We added 2 supplementary videos (video S4 and S5) of NMJ inhibition and descript the result at line 237-240.
- Images of the staining of the misfolded proteins in figure 6 should be improved by providing higher magnification figures.
Reply: We are sorry for the unclear images. The current magnification is the limitation of our microscope. Thus, we thawed the SOD1G85R and sporadic ALS MN stocks and co-stain misfolded SOD1 with β-III-tubulin (figure 6e and 6g) for clear vision. The dot like SOD1 accumulations in cell soma are indicated with arrows. We also try to improve the TDP-43 presentation in figure 6 with separate images that may be helpful to observe the cell nucleus and TDP-43 individually (figure 6c and 6d). In the merged image (c3 and d3), we label the nucleus shape with red circles, and TDP-43 redistributions are indicated by arrowheads. If the figures integrated in the word or PDF file are not clear enough, please refer to the individual figure JPEG files for high resolution image.
- English language can be improved.
Reply: Thank you for your suggestion. This article was edited by whose native language is English before submission. In this revision, we do the second time English editing then carefully modify the article by the authors, modifications are majorly at line 44, 60, 67, 101-102, 121-122, 133, 254-259, 465-470. We hope these will improve our English presentation.

Round 2
Reviewer 1 Report
Major:
Fig 1 is impossible to understand.
The quantification Figures 1q and 1r are inconsistent with the images and the text.
In images, NF and Isl1 markers appear to be lower in the new (CHSF) method compared to the old method (including in Fig. 1r). Isn’t it supposed to be just the opposite, according to the manuscript’s idea? NF marker was not quantified.
In images, HB9 marker appears to be higher in the new method, but in Fig. 1r it is lower.
The quantifications in Fig. 1 are inconsistent with those in Fig. 2.
Fig. 1q and 1r both contain NKX2.2 marker (why?), but show opposite ratios between the two methods.
Overall, it’s a total mess.
Images in Fig. 2 have no indication at which time point they were taken. Fig. 2s is a mixture of the markers on day 15 and day 25. They should be separated to different graphs.
Line 163-164: “On day 25 of differentiation, 91.34±4.26%–93.54±1.34% HB9 was elevated in all PSC lines (Figure 2s).” What this sentence is supposed to mean?
It looks like the scale bars should be different in the images in Fig. 2, but the legend to Fig. 2 states they are the same.
No error bar in Supplementary Fig. S1.
L. 401-405. There is no explanation how the cell boundaries were defined to enable cell counting. It's crucial, since all the quantifications are represented by a marker+/DAPI+ cell ratio. It's very difficult to guess individual cells in the majority of the images presented.
All the new additions to the manuscript's text were made in a hurry and written in a very bad English. Just one example: Lines 296-308. The entire manuscript, especially new additions, must undergo a professional scientific English editing.
Author Response
Dear Editor:
Thank you very much for your appreciation and reviewers’ suggestions for our manuscript ‘Robust Generation of Ready-to-Use Cryopreserved Motor Neurons from Human Pluripotent Stem Cells for Disease Modeling’. We are very grateful for the opportunity for improving our works. According to the recommendations of reviewers, we improve the presentation of figures, modify the article content, redo the scientific English language editing, and response to the suggestions point by point in the reply letter and highlight the changes with “Track Changes” and “Yellow Highlight Color” in the revised text. We sincerely hope this revised manuscript will be given a serious consideration for publishing in International Journal of Molecular Sciences.
Sincerely yours,
Chia-Yu Chang
Bioinnovation Center, Buddhist Tzu Chi Medical Foundation, Hualien, Taiwan.
No. 707, Sec. 3, Chung-Yang Rd. Hualien, Taiwan, 97002
Phone No: +886-3-856-1825 (ext. 12106)
Email Address: scata0726@hotmail.com
Reply Letter to Reviewer 1
Comments and Suggestions for Authors
Major:
Fig 1 is impossible to understand.
The quantification Figures 1q and 1r are inconsistent with the images and the text.
Reply: I’m sorry that we make the mistake on the presentation of Data set in Figure 1r. The white color bars are BiSF original method, and the black bars are CHSF-MN method, but we labeled them with opposite colors in the previous version Figure 1r. We correct the Figure 1r with the right color symbols.
In images, NF and Isl1 markers appear to be lower in the new (CHSF) method compared to the old method (including in Fig. 1r). Isn’t it supposed to be just the opposite, according to the manuscript’s idea? NF marker was not quantified.
Reply: In addition to the wrong symbols in Figure 1r, reader may feel like the NF expression in Figure 1g (BiSF original) is more than Figure 1o (CHSF-MN) because of there are large, aggregated cell clumps with more cells in the visual field of 1g. To avoid this possibility of misunderstanding, we change the Figure 1g with an image that have less cells and without large, aggregated cell clumps. We calculate the day 25 NF expression of BiSF (84.57±4.92%) and CHSF-MN (98.41±0.21%), present in Figure 1r and describe at line 116 and line 146. We also calculate the NF expression of each line in Figure 2t and describe at Line 185.
The Oligo2 and Islet1 are both MN related markers in the ventral-posterior spinal cord. However, Oligo2 would express at the progenitor stage (early) and the Islet1 express at pre-mature neuron stage (late). Thus, these two markers would not co-express at the same differentiation stage. They express sequentially[1-3]. At day 15 of differentiation (progenitor stage), the major expression stage of Oligo2, there are very few islet1 expression in both BiSF (Figure 1b) and CHSF-MN (Figure 1j). At day 25 of differentiation (pre-mature neuron stage), the Islet1+HB9+ cells indicate the pre-mature MN. At this stage, the Islet1 expression in CHSF-MN (96.48±0.66%; Figure 1n and 1r) is much more than BiSF (55.74±31.21%, about half of DAPI+ cells are lack of Islet1 and HB9 expression; Figure 1f and 1r). The data is calculated in the corrected Figure 1r and describe at line 118, line 152 and line 154-160.
The Figure 1q calculate the marker expression data of day 15 and Figure 1r calculate the day 25 data. We add the information in the result and figure legend at line 163-164 and line 175. Sorry for the lack of information.
In images, HB9 marker appears to be higher in the new method, but in Fig. 1r it is lower.
Reply: We are sorry for the wrong symbols in Figure 1r and correct it.
The quantifications in Fig. 1 are inconsistent with those in Fig. 2.
Reply: This is also due to the opposite color symbols in Figure 1r. Sorry for this mistake.
Fig. 1q and 1r both contain NKX2.2 marker (why?), but show opposite ratios between the two methods.
Reply: The Figure 1q is day 15 data and 1r is day 25, we add the information in the result and figure legend at line 163-164 and line 175. The mistake we made in the Figure 1r is also lead to confusion. The NKX2.2 is a ventral-posterior spinal cord marker that expressed in the common pool of Oligo2+ neural progenitor that can finally become interneurons (NKX2.2+ but HB9-) or MNs (HB9+ but NKX2.2-). Thus, we track the NKX2.2 expression at day 15 and day 25 to indicate the final fate of these neural progenitors. In the BiSF protocol, most of neural progenitors are NKX2.2+ at day 15 (87.45±4.81%; Figure 1q), and finally about half of them become NKX2.2+ interneurons at day 25 (52.44±4.12%; Figure 1r). In the CHSF-MN protocol, most NKX2.2 are inhibited at day 15 (7.05±2.13%; Figure 1q) and almost undetectable at day 25 (Figure 1r), indicate the NKX2.2 inhibition may promote neural progenitors to become MNs. The role of NKX2.2 is added at line 120-125 and line 160-163.
Overall, it’s a total mess.
Reply: We are sorry for the wrong symbols in Figure 1r.
Imags in Fig. 2 have no indication at which time point they were taken. Fig. 2s is a mixture of the markers on day 15 and day 25. They should be separated to different graphs.
Reply: Thank you for your suggestions. We add the ICC timing information beside the name of each cell line in Figure 2. The Figure 2s is separated into Figure 2s (day 15) and 2t (day 25). The time information is added in the result and figure legend at Line 186 and 194-195.
Line 163-164: “On day 25 of differentiation, 91.34±4.26%–93.54±1.34% HB9 was elevated in all PSC lines (Figure 2s).” What this sentence is supposed to mean?
Reply: We correct this sentence as “On day 25, the differentiated cells expressed 96.92% ± 1.03% to 99.23% ± 0.19% NF and 91.34% ± 4.26% to 93.54% ± 1.34% HB9 in all PSC lines” at line 185-186.
It looks like the scale bars should be different in the images in Fig. 2, but the legend to Fig. 2 states they are the same.
Reply: We examine the image record of all original files that presented in Figure 2 and confirm that they are captured under 200X magnification. The cell clump size in individual image and the cell morphological difference between day 15 and day 25 may cause the visual effect and feel like the images are under different magnification.
No error bar in Supplementary Fig. S1.
Reply: sorry for the lack of error bar. We add the error bar on Figure s1 and describe the ChAT expression ratio at line 232-234.
- 401-405. There is no explanation how the cell boundaries were defined to enable cell counting. It's crucial, since all the quantifications are represented by a marker+/DAPI+ cell ratio. It's very difficult to guess individual cells in the majority of the images presented.
Reply: We calculate the DAPI+ cell nucleus that without overlapping with other ones, and can be clearly identified under 400X magnification, for quantification. We add this information at line 434-436. Most of images in Figure 1 and Figure 2 are 200X magnification, to show some specific structures, such as the Rosette like neural stem cells, and for presenting an overall view of marker expression.
All the new additions to the manuscript's text were made in a hurry and written in a very bad English. Just one example: Lines 296-308. The entire manuscript, especially new additions, must undergo a professional scientific English editing.
Reply: Thank you for providing us the opportunity to improve the English presentation. This article was edited by whose native language is English before submission, but we are sorry the language quality is poor. This time we carefully examine the full article and rewrite some parts (especially the new added parts) for improving the expression, and redo the full manuscript English editing. Since there are numerous modifications while full manuscript English editing, we do not highlight all minor modified points. The major modifications are labeled with “Track Changes” and the major rewrite or new added parts are labeled with “Track Changes + Yellow Highlight Color” at line 120-125, 154-164, 230-235, 259-262, 284-288, and 326-338. Hope our effort could improve the quality of manuscript.
References:
- Maury, Y.; Come, J.; Piskorowski, R.A.; Salah-Mohellibi, N.; Chevaleyre, V.; Peschanski, M.; Martinat, C.; Nedelec, S. Combinatorial analysis of developmental cues efficiently converts human pluripotent stem cells into multiple neuronal subtypes. Nat Biotechnol 2015, 33, 89-96, doi:10.1038/nbt.3049.
- Du, Z.W.; Chen, H.; Liu, H.; Lu, J.; Qian, K.; Huang, C.L.; Zhong, X.; Fan, F.; Zhang, S.C. Generation and expansion of highly pure motor neuron progenitors from human pluripotent stem cells. Nat Commun 2015, 6, 6626, doi:10.1038/ncomms7626.
- Li, X.J.; Du, Z.W.; Zarnowska, E.D.; Pankratz, M.; Hansen, L.O.; Pearce, R.A.; Zhang, S.C. Specification of motoneurons from human embryonic stem cells. Nat Biotechnol 2005, 23, 215-221, doi:10.1038/nbt1063.

Round 3
Reviewer 1 Report
line 185-186: “On day 25, the differentiated cells expressed 96.92% ± 1.03% to 99.23% ± 0.19% NF and 91.34% ± 4.26% to 93.54% ± 1.34% HB9 in all PSC lines”
This is still wrong. It should be:
"On day 25, 96.92% ± 1.03% to 99.23% ± 0.19% of the differentiated cells expressed NF and 91.34% ± 4.26% to 93.54% ± 1.34% expressed HB9 in all PSC lines."
Author Response
Dear Editor:
Thank you very much for your appreciation and reviewers’ suggestion for our manuscript ‘Robust Generation of Ready-to-Use Cryopreserved Motor Neurons from Human Pluripotent Stem Cells for Disease Modeling’. We are very grateful for the opportunity for improving our works. According to the recommendation of reviewer, we modify the specific sentence at line 185-186 and response to the suggestion point in the reply letter and highlight the change with “Track Changes” in the revised text. We sincerely hope this revised manuscript will be given a serious consideration for publishing in International Journal of Molecular Sciences.
Sincerely yours,
Chia-Yu Chang
Bioinnovation Center, Buddhist Tzu Chi Medical Foundation, Hualien, Taiwan.
No. 707, Sec. 3, Chung-Yang Rd. Hualien, Taiwan, 97002
Phone No: +886-3-856-1825 (ext. 12106)
Email Address: scata0726@hotmail.com
Reply Letter to Reviewer 1
Comments and Suggestions for Authors
line 185-186: “On day 25, the differentiated cells expressed 96.92% ± 1.03% to 99.23% ± 0.19% NF and 91.34% ± 4.26% to 93.54% ± 1.34% HB9 in all PSC lines”
This is still wrong. It should be:
"On day 25, 96.92% ± 1.03% to 99.23% ± 0.19% of the differentiated cells expressed NF and 91.34% ± 4.26% to 93.54% ± 1.34% expressed HB9 in all PSC lines."
Reply: Thank you to help us improving our English expression quality. This sentence is modified as “On day 25, 96.92% ± 1.03% to 99.23% ± 0.19% of the differentiated cells expressed NF and 91.34% ± 4.26% to 93.54% ± 1.34% expressed HB9 in all PSC lines.” at line 185-186.
